# Whole Transcriptome Analysis of Intervention Effect of *Sophora subprostrate* Polysaccharide on Inflammation in PCV2 Infected Murine Splenic Lymphocytes

Yi Zhao [†][ID], Nina Jia [†], Xiaodong Xie [ID], Qi Chen and Tingjun Hu *

College of Animal Science and Technology, Guangxi University, Nanning 530005, China;
zhaoyi@st.gxu.edu.cn (Y.Z.); ninajia@st.gxu.edu.cn (N.J.); 1718304007@st.gxu.edu.cn (X.X.);
chenqi@st.gxu.edu.cn (Q.C.)
* Correspondence: tingjunhu@gxu.edu.cn; Tel.: +86-771-3235635; Fax: +86-771-3270149
† These authors contribute equally to this work.

**Abstract:** (1) Background: *Sophora subprostrate*, is the dried root and rhizome of *Sophora tonkinensis* Gagnep. *Sophora subprostrate* polysaccharide (SSP1) was extracted from *Sophora subprostrate*, which has shown good anti-inflammatory and antioxidant effects. Previous studies showed SSP1 could modulate inflammatory damage induced by porcine circovirus type 2 (PCV2) in murine splenic lymphocytes, but the specific regulatory mechanism is unclear. (2) Methods: Whole transcriptome analysis was used to characterize the differentially expressed mRNA, lncRNA, and miRNA in PCV2-infected cells and SSP1-treated infected cells. Gene Ontology (GO), Kyoto Encyclopedia of Genes and Genomes (KEGG) and other analyses were used to screen for key inflammation-related differentially expressed genes. The sequencing results were verified by RT-qPCR, and western blot was used to verify the key protein in main enriched signal pathways. (3) Results: SSP1 can regulate inflammation-related gene changes induced by PCV2, and its interventional mechanism is mainly involved in the key differential miRNA including miR-7032-y, miR-328-y, and miR-484-z. These inflammation-related genes were mainly enriched in the TNF signal pathway and NF-κB signal pathway, and SSP1 could significantly inhibit the protein expression levels of p-IκB, p-p65, TNF-α, IRF1, GBP2 and p-SAMHD1 to alleviate inflammatory damage. (4) Conclusions: The mechanism of SSP1 regulating PCV2-induced murine splenic lymphocyte inflammation was explored from a whole transcriptome perspective, which provides a theoretical basis for the practical application of SSP1.

**Keywords:** inflammation; murine splenic lymphocyte; porcine circovirus type 2 (PCV2); *Sophora subprostrate* polysaccharide (SSP1); whole transcriptome analysis

## 1. Introduction

Porcine circovirus type 2 (PCV2) is a single-stranded, non-encapsulated circular DNA virus that causes porcine circovirus-related diseases and is one of the most important pathogens of swine disease in the world [1]. The pathogenesis of PCV2 has not been thoroughly studied, but a large number of studies have shown that the occurrence of the disease may be related to the body's immune system being attacked by viruses. The natural hosts of PCV2 are pigs, and a large number of PCV2 virus particles can be detected in the lymphoid tissue, lungs and spleen of PCV2-infected pigs [2]. Research has shown that PCV2 infection can activate or inhibit MAPK, NF-κB, PI3K/AKT and other signal pathways to initiate inflammation in the host. It also leads to host lymphatic depletion and a decrease in the number of B cells and T cells in immune organs causing immunosuppression of pigs [3]. Previous studies have shown that PCV2 can up-regulate the phosphorylation level of IκBα protein and activate phosphorylated NF-κB p65 into the nucleus [4], while inhibiting NF-κB activation can effectively reduce the replication efficiency of PCV2 in the host [5]. PCV2 can inhibit apoptosis, improve the survival rate of infected cells, and promote viral replication

by activating the PI3K/AKT pathway [6]. However, the mechanism of PCV2-induced inflammatory response and related signaling pathways is unclear.

Natural Chinese herbal medicines have been found to have good effects in the prevention and treatment of virus infection, not only as immune enhancers to improve the body's disease resistance but also to resist viral infection by inhibiting virus replication and apoptosis and reducing the inflammatory response. Matrine plays an anti-infective role by inhibiting PCV2 replication in the liver of mice with mixed PRRSV/PCV2 infection and promoting lymphocyte proliferation and macrophage activation [7]. *Astragalus* polysaccharides significantly reduce PCV2 infection by inhibiting endoplasmic reticulum stress responses in PK-15 cells [8]. *Sophora subprostrate* polysaccharide is one of the main bioactive components of Chinese herbal *Sophora subprostrate*, with a variety of biological activities such as anti-inflammation, antioxidant and immunomodulation [9–11]. Previous studies have found that *Sophora subprostrate* polysaccharides can inhibit the inflammatory response of splenic lymphocytes induced by PCV2 infection by regulating the expression of histone acetylation-related genes and proteins [12], but the mechanisms by which *Sophora subprostrate* polysaccharides regulate the inflammatory response of murine splenic lymphocytes is still unclear, especially the specific way the *Sophora subprostrate* polysaccharide regulates the inflammatory signaling pathways still needs to be explored.

Long noncoding RNAs (LncRNAs) are a class of endogenous cellular RNAs that lack an open reading coding frame, similar to mRNA, and can interact with DNA, RNA, and proteins to regulate transcription, epigenetic modifications, protein/RNA stability, translation, and post-translational modifications, or directly with signaling receptors [13,14]. Many lncRNAs play key roles in various physiological and pathological processes, including tumor formation and metastasis, inflammation, apoptosis, differentiation, autophagy and metabolism [15,16]. Much research showed that identifying and analyzing the role of non-coding RNAs is an important way to understand the core genes of the regulatory pathways of biologically active substances [17,18]. The function of lncRNAs is typically mediated by regulating microRNAs (miRNAs), which regulate gene expression by binding to the 3′ untranslated region (UTR) of mRNA. miRNAs are a class of small, non-coding RNAs containing about 20 nucleotides that control cellular processes by regulating gene expression after transcription [19,20]. miRNA can regulate the expression level of mRNA by binding to the complementary sequence of the target mRNA, thereby regulating biological processes [21]. Dysregulation of miRNAs can lead to the occurrence of inflammation, immunity, tumors and cardiovascular diseases [22].

LncRNA and miRNA are involved in the regulation of mRNA levels, thereby regulating protein synthesis in many reports. NF-κB is a key inflammatory pathway that connects chronic inflammation and cancer transformation, and several studies have shown that lncRNAs directly or indirectly regulate the NF-κB signaling pathway, promoting disease development and progression. MicroRNAs and lncRNAs are emerging as another key point in the complex regulatory structure that controls the NF-κB signaling pathway. Knocking down LncRNA H19 expression can reduce the phosphorylation expression of phosphoinositide 3 kinase (PI3K), protein kinase B (Akt) and IκBα, prevent the migration and invasion of melanoma cells from the nuclear translocation of p-p65 and activate the NF-κB signaling pathway [23,24]. Recent studies have shown that lncRNA and miRNA are closely related to the PCV2 infection process. PCV2 can dysregulate the miRNA-mRNA regulatory network in PK-15 cells and induce inflammatory responses, overexpression or inhibition of miRNA expression (miR-10b, miR-128, miR-155-5p, miR-21, etc.), and can regulate the PCV2-induced inflammatory responses [25]. PCV2 regulates the HOXB gene by regulating lncRNAs in PK-15 cells, which may be key to influencing embryonic development [26]. However, whether SSP1 can exert its biological activity by regulating the LncRNA-miRNA-mRNA axis still needs to be explored. This study aimed to evaluate the potential of SSP1 in preventing PCV2-induced inflammatory damage of murine splenic lymphocytes, detect differentially expressed lncRNA (DE lncRNA), miRNA (DE miRNA)

and mRNA by RNA-seq, and construct a related network to explain the mechanism of action of SSP1 and provide new ideas for the prevention and treatment of PCV2.

## 2. Materials and Methods

### 2.1. SSP1 and Virus

SSP1 was separated using DEAE cellulose ion exchange chromatography and purified by Sephadex G 100 (Pharmacia, Shanghai, China) chromatography. The structure of SSP1 was analyzed through periodate oxidation analysis, smith degradation, methylation analysis, IR, $^1$H and $^{13}$C NMR analysis. SSP1 is a (1→4) linked α glucan to which are attached two glucosyl side chains at 3-O and 6-O of the glucosyl residues in every 12 repeating units of the main chain [11] (Figure 1). The final concentration of SSP1 used in this experiment used 400 μg/mL with a purity of 99.9% and less than 0.005 ng/mg endotoxin.

**Figure 1.** The structure of SSP1.

PCV2 (SH strain) and PCV2 (PCV2NJ2002) were maintained by the Pharmacology Laboratory, College of Animal Science and Technology, Guangxi University, proliferated in PK-15 cells, and stored at −80 °C. The PCV2 strain (PCV2NJ2002) was isolated from a kidney tissue sample of a pig with naturally occurring postweaning multisystemic wasting syndrome (PMWS), and identified by sequencing (Invitrogen, Carlsbad, CA, USA). PCV2 was passaged seven times in PK-15 cells and the TCID$_{50}$ of these two strains was calculated to be $10^{-5}$/0.1 mL by the Reed-Muench method, respectively. PK-15 cell was provided by the China Center for Type Culture Collection (Number: GDC0061).

### 2.2. Isolation and Treatment of Murine Splenic Lymphocytes

SPF Kunming mice (20.0 ± 2.0 g) were purchased from the experimental animal center of Guangxi Medical University. After 3 days of feeding, the mice were euthanized and dissected and spleens were harvested in a sterile environment. Murine splenic lymphocytes were obtained in accordance with the instructions of the murine spleen lymphocyte isolate kit. The cells were cultured with RPMI 1640 medium culture with 10% FBS, 100 U/mL penicillin and 0.1 mg/mL streptomycin. The CD4+ and CD8+ purity of lymphocytes was assessed by flow cytometry. Lymphocyte suspensions containing CD4+ and CD8+ fluorescently labeled antibodies were added to the Flowjo tubes and repeated three times. The CD4+ cells accounted for 38.12 ± 2.01 (%) and the CD8+ cells for 10.3 ± 0.59 (%) using Flowjo software (BD Biosciences, Franklin Lakes, NJ, USA).

### 2.3. Reagents

The murine splenic lymphocyte isolation liquid kit was purchased from Thermo Fisher Scientific (Waltham, MA, USA). RPMI-1640 medium culture and fetal bovine serum (FBS) were purchased from Gibco (Gibco, Grand Island, NY, USA). The streptomycin penicillin mixed solution and murine spleen lymphocyte isolate kit were purchased from Solarbio (Beijing Solarbio Technology Co., Ltd., Beijing, China). RNAiso Plus was purchased from Takara (Takara Biomedical Technology Co., Ltd., Beijing, China). All-In-One 5× RT MasterMix and BlasTaq$_{TM}$ 2× qPCR MasterMix were purchased from ABM (Applied Biological Materials Inc., Richmond, BC, Canada). CD4 Monoclonal Antibody and CD8a Monoclonal Antibody were purchased from Thermo Fisher Scientific (Waltham, MA, USA).

The RIPA lysis buffer, protease inhibitor, phosphatase inhibitor and BCA protein assay kit were purchased from Beyotime (Beyotime Biotechnology Co., Ltd., Shanghai, China). The PVDF membrane was purchased from Roche (F. Hoffmann-La Roche Ltd., Basel, Switzerland). NF-κB p65 rabbit mAb, phospho-NF-κB p65 rabbit mAb, IkBα rabbit mAb, phospho-IκBα rabbit mAb, TNF-α rabbit mAb, β-actin antibody and anti-rabbit IgG HRP-linked antibody were purchased from Abcam (Abcam Plc., Cambridge, UK); IRF1 (D5E4) Rabbit mAb was purchased from CST (Cell Signaling Technology, Inc., Boston, MA, USA); GBP2 Rabbit pAb and SAMHD1 Rabbit pAb were purchased from Abclonal (ABclonal Technology Co., Ltd., Wuhan, China); Phospho-SAMHD1 (Thr592) Antibody was purchased from Abmart (Abmart Shanghai Co., Ltd., Shanghai, China). Immobilon ECL Ultra Western HRP Substrate was purchased from NCM Biotech (New Cell & Molecular Biotech Co., Ltd., Suzhou, China). The reagents 30% acrylamide, ammonium persulfate, 10% SDS, 1 M Tris HCl (pH 6.8), 1.5 M Tris HCl (pH 8.8), skim-milk, and TBST (pH 8.0, 10×) were all purchased from Solarbio (Beijing Solarbio Technology Co., Ltd., Beijing, China).

*2.4. Cell Grouping and Treatment*

In our previous research, both the PCV2 SH strain and PCV2NJ2002 were used to explore the inflammation-causing effect in murine spleen lymphocytes. The levels of inflammatory response in PCV2-infected murine spleen lymphocytes were significantly increased, and SSP1 could alleviate PCV2-induced inflammatory injury [12,27]. PCV2 SH strain [12] and PCV2NJ2002 [27] showed similar effects during the experiment, and SSP1 treatment at 400 μg/mL showed good anti-inflammatory effects. To investigate the effect of SSP1 on PCV2-infected cells on gene regulation, the following three groups were set up: Control (C), PCV2 infection group (PCV2, V), and SSP1 treatment group (PCV2 + 400 μg/mL SSP1, SV). Murine splenic lymphocytes were diluted with 5% FBS-RPMI-1640 to $1 \times 10^6$ cells/mL, then added to the 6-well plate with 2 mL, and cultured in the 37 °C incubator of 5%CO$_2$ for 6 h. The supernatant was discarded, and the cells were inoculated with PCV2 (MOI = 1) for 2 h, except for the control groups. After discarding the supernatant, the RPMI-1640 was added in C and V group, and the cells in the SV group were treated with 400 μg/mL SSP1 and further cultured for 24 h.

*2.5. RNA Sequencing and Bioinformatics Analysis*

The total RNA was extracted strictly according to the instructions of the RNAiso Plus. Then the purity, quantity, and integrity of extracted RNA were measured by agarose gel electrophoresis, Nanodrop microspectrophotometer detection (Thermo Fisher Scientific, Waltham, MA, USA), and Agilent2100 detection (Agilent Technologies, Palo Alto, CA, USA). After obtaining the total cellular RNA, use the Illumina Ribo-Zero (PlantLeaf) kit to remove the rRNA, preserving all coding RNA and ncRNA as much as possible. RNA is randomly interrupted to obtain short RNA fragments, which are used as templates to synthesize the first strand of cDNA using random hexamers. Based on the first strand, buffer, dNTPs, RNaseH, and DNaseI (Thermo Fisher Scientific, Waltham, MA, USA) were added, and the second strand of cDNA was synthesized in a PCR after thorough mixing. The obtained second strand of cDNA was purified using 1.8× Agencourt AMPure XP Beads, and the purified cDNA was pipetted for end repair. The purified double-stranded cDNA fragments were end-repaired and a base was added and ligated to Illumina sequencing adapters by using NEBNext Ultra RNA Library Prep Kit (NEB #7530, New England Biolabs, Ipswich, MA, USA). The above products were recovered and degraded by the Uracil-N-glycosylase enzyme (Thermo Fisher Scientific, Waltham, MA, USA), selected by agarose gel electrophoresis, and amplified by PCR. The PCR reaction procedure was as follows: NEBNext USER Enzyme, 3 μL; NEBNext High-Fidelity PCR Master Mix, 2×, 25 μL; Universal PCR Primer (25 mM), 1μL; Index (X) Primer (25 mM), 1 μL; reaction solution, 22 μL; 98 °C, 30 s; (98 °C, 10 s; 65 °C, 75 s) for 12 cycles; 65 °C, 5 s. For amplification products, Illumina novaseq 6000 sequencing (Gene Denovo Biotechnology Co., Ltd., Guangzhou, China) was used.

We filtered the raw reads after disembarking to reduce invalid data and finally obtained clean reads. Clean reads were compared to the ribosome database of mice by bowtie, and the reads compared to the ribosome were removed, and the remaining part was unmapped reads. The mouse reference genome was compared using HISAT2 software and the transcript was reconstructed by String tie software (The Center for Computational Biology at Johns Hopkins University, Baltimore, MD, USA).

### 2.6. Differential Expression Analysis

The expression profiles of each LncRNA, mRNA, and miRNA were quantified and normalized by the DEseq method, and the differential expressed genes among each group were calculated by *p*-value. The significant DEGs were recognized with an adjusted *p*-value $\leq 0.05$ and a $\geq 1.2$-fold change in expression. The DEGs were enriched by their Gene Ontology (GO) or Kyoto Encyclopedia of Genes and Genomes (KEGG) pathways with the GSEA method, as well as the lncRNA annotation, Encyclopedia of DNA Elements (ENCODE) database, and mapping to Reactome database using the cluster profiler package.

### 2.7. Trend Analysis of mRNAs and Transcription Factor Targeting Analysis

With reference to STEM software (Gene Denovo Biotechnology Co., Ltd., Guangzhou, China), log2 standardization was used to preprocess gene expression and cluster trend analysis was conducted on the change trend of genes among samples in each group.

Motif information of transcription factor binding was obtained using the JASPER database (https://jaspar.genereg.net/about/ (accessed on 15 September 2022)), and MEME FIMO software was used to predict transcription target genes. This data analysis was processed in Omicsmart (Gene Denovo Biotechnology Co., Ltd., Guangzhou, China).

### 2.8. RT- qPCR to Detect Differentially Expressed RNA

Differentially expressed mRNAs, LncRNAs, and miRNAs were randomly chosen for fluorescence quantitative PCR verification (RT-qPCR) to confirm the reliability of the RNA-seq data. The primers for mRNA, lncRNA, and miRNA were synthesized by Sangon (Sangon Biotech. Co., Ltd., Shanghai, China). After cDNA was obtained by reverse transcription, the relative mRNA, lncRNA, and miRNA content was quantified by real-time PCR. The reaction system is as follows: BlasTaq TM $2\times$ qPCR MasterMix, 10 μL; cDNA, 1 μL; forward primer (10 μM), 0.5 μL; reverse primer (10 μM), 0.5 μL; Nuclease free $H_2O$, 8 μL. PCR reaction procedure was as follows: Pre-degeneration, 95 °C, 180 s; degeneration, 95 °C, 15 s; extension, 60 °C, 60 s, 40 cycles; melting curves were performed according to the instrument's own settings. The primers listed in Tables S3–S5 were used for PCR amplification. For miRNA, Gene expression levels were calculated by normalizing the CT values to β-actin mRNA, and data analysis was conducted using the $2^{-\Delta\Delta CT}$ comparative threshold method. The RT-qPCR machine used in this experiment was LightCycler® 96 Instrument (Roche, Basel, Switzerland).

### 2.9. Western Blot

After 24 h of treatment, total protein was isolated from cells by RIPA lysate with protease inhibitor and phosphatase inhibitor. After determining the protein concentration by the BCA protein assay kit, 20 μg protein was added into each well of the vertical electrophoresis tank and separated by 10% SDS-PAGE. Subsequently, the protein was transferred onto the PVDF membrane. The Power Pac basic power supply, Wide mini-sub cell GT system, Trans-Blot SD Semi-Dry electrophoretic transfer tank were all provided by Bio-Rad (Bio-Rad Laboratories, Inc., Hercules, CA, USA). After being blocked by 5% skim-milk for 2 h, then the incubation of primary antibodies (TNF-α, p65, p-p65, IκBα, p-IκBα, GBP2, IRF1, SAMHD1, p-SAMHD1, β-actin, GAPDH, 1:2000) was performed overnight at 4 °C. After washing by TBST three times for 30 min, the secondary antibodies (anti-rabbit IgG HRP-linked antibody, 1:2000) were incubated for 1 h at room temperature. After TBST wash, the ECL chromogenic substrate was used to detect specific bands in

Image Quant LAS 500 (GE HealthCare Technologies Inc., Chicago, IL, USA). The gray value was estimated by Image J software (National Institutes of Health, Bethesda, MD, USA) normalized to β-actin.

### 2.10. Target Gene Prediction and ceRNA Network Construction

To explore the genetic changes of SSP1 in regulating PCV2 infection, differentially expressed inflammatory genes were determined by |Fold Change| ≥ 1.2 and $p < 0.05$. miRNAs negatively correlated with differential mRNA and LncRNAs negatively correlated with miRNA were screened using the miRnada analysis tool. The interactions of miRNA-LncRNA, and miRNA-mRNA were integrated to construct the competing endogenous RNA (ceRNA) regulatory network using Cytoscape version 3.6.0 (National Resource for Network Biology).

### 2.11. The Interaction between LncRNAs, miRNAs, and mRNAs in V Group and S Group

The mRNAs related to inflammatory response were screened from 158 differentially co-expressed mRNAs between the C group, V group, and SV group, and the differential lncRNA with trans, cis, and antisense relationships and differential miRNAs with targeted relationships with mRNA were screened, and the LncRNA-miRNA-mRNA network diagram was constructed by Cytoscape version 3.6.0.

### 2.12. Statistical Analysis

The data was analyzed by SPSS 21.0 software (SPSS, Chicago, IL, USA). The differences in groups were estimated by One-way ANOVA analysis. The results are presented as the mean ± SD. $p < 0.05$ or $p < 0.01$ meant that the difference was significant or extremely significant.

## 3. Results

### 3.1. Effect of SSP1 on the mRNA, lncRNA, and miRNA Expression of the PCV2-Infected Murine Splenic Lymphocytes

To further study the regulatory effect of SSP1 on the inflammatory response of PCV2-infected murine splenic lymphocytes, the differential expression of mRNA, lncRNA, miRNA, and circRNA was detected by RNA-seq. In this study, about 110853910 to 82165908 raw reads and 110336690 to 81768490 clear reads per sample were obtained, then an index of the reference genome was built, and paired-end clean reads were mapped to the reference genome (Ensembl-release 104) using HISAT2. 2.4 and other parameters set as a default (Table S1). RNA differential expression analysis was performed by DESeq2 software between two different groups. The genes/transcripts with the parameter of $p$ value ($p$) below 0.05 and absolute fold change ≥ 1.2 were considered differentially expressed genes/transcripts. The results showed that there were 1607 mRNAs, 549 lncRNAs, 144 miRNAs and 21 circRNAs differentially expressed between the C group and the V group. PCV2 infection significantly up-regulated 894 mRNAs, 293 lncRNAs, 85 miRNAs, and 15 circRNAs and down-regulated 713 mRNAs, 256 lncRNAs, 59 miRNAs, and six circRNAs (Figures 2A,F and 3A,F). Compared with the V group, 303 mRNAs, 163 lncRNAs, 37 miRNAs, and 12 circRNAs were up-regulated, while 327 mRNAs, 194 lncRNAs, 80 miRNAs, and 11 circRNAs were down-regulated in SV groups (Figures 2C,H and 3C,H). The most differentially expressed top 20 genes in each group were shown in the heat maps (Figures 2B,D,G,I and 3B,D,G,I). A Venn diagram was constructed to show the common differentially expressed genes in the three groups (Figures 2E,J and 3E,J). There were 158 mRNAs differentially expressed among C, V, and SV groups, and 19 mRNAs are related to inflammatory responses (Figure 4).

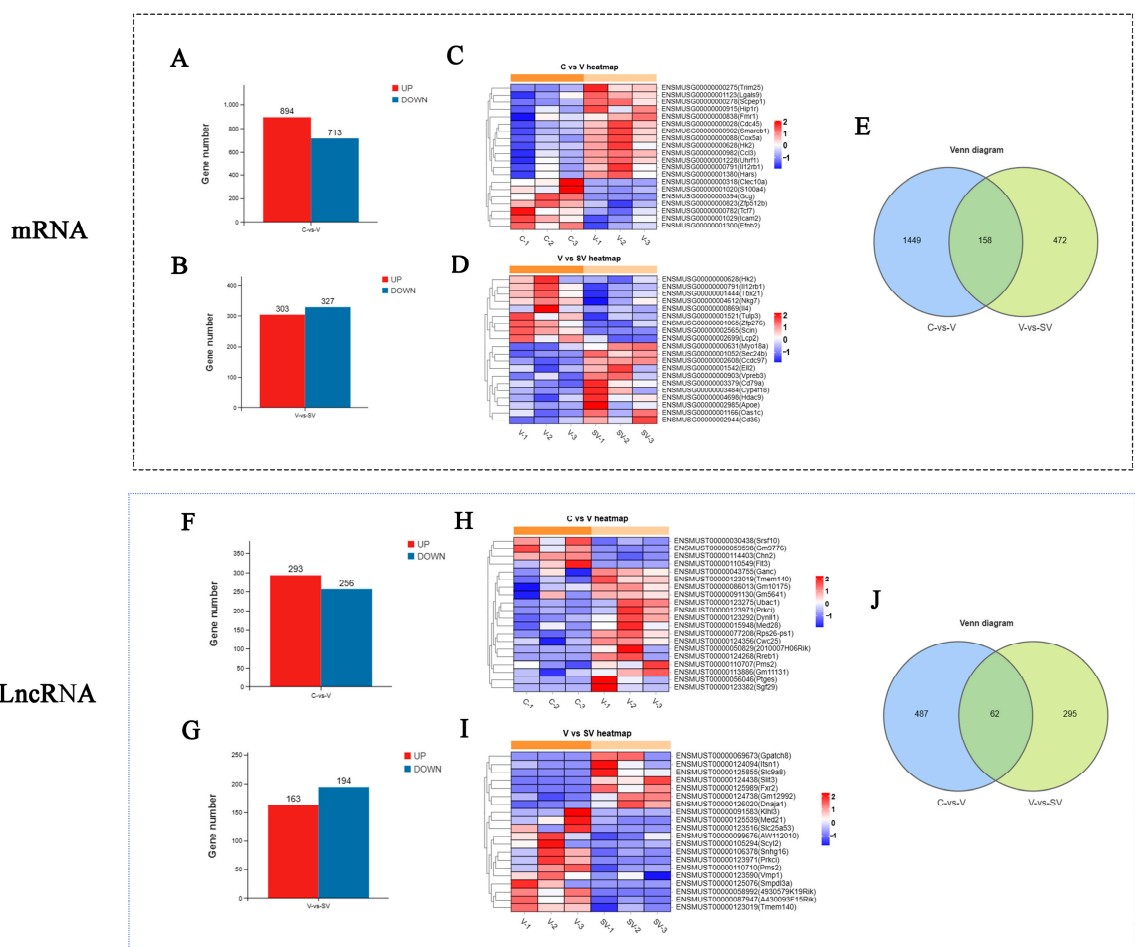

**Figure 2.** Histogram, cluster heat map and Venn chart of mRNA and LncRNA. 1607 differentially expressed mRNAs (**A**) and 549 lncRNAs (**F**) between C group and V group (FC ≥ 1.2, *p* < 0.05). 630 differentially expressed mRNAs (**C**) and 357 lncRNAs (**H**) between V group and SV group (FC ≥ 1.2, *p* < 0.05). Heat map generated by hierarchical clustering analysis of differentially expressed top 20 mRNAs (**B**,**D**) lncRNAs (**G**,**I**). 158 differentially co-expressed mRNAs (**E**) and 62 lncRNAs (**J**) were shown in Venn chart.

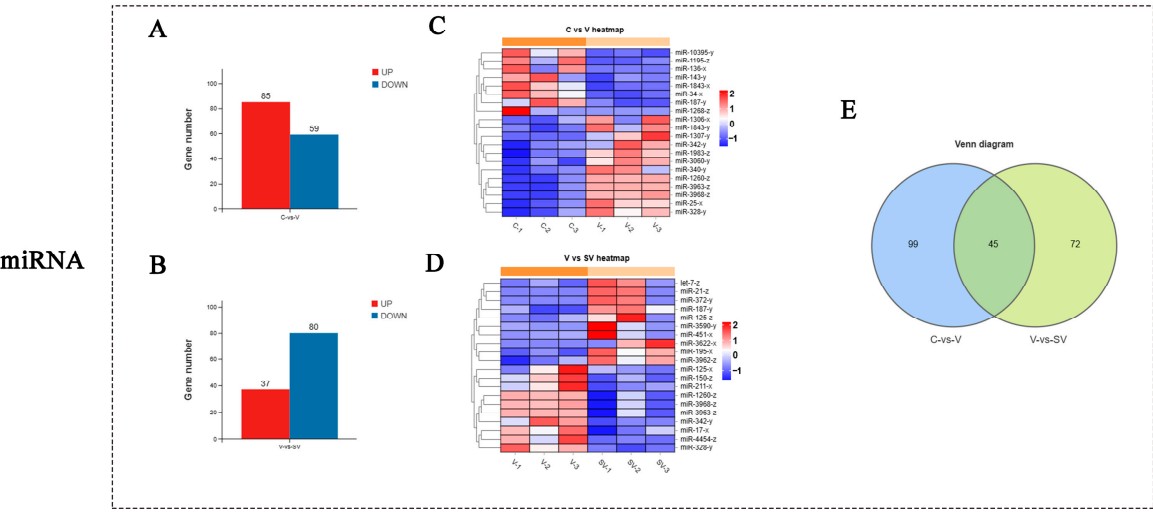

**Figure 3.** *Cont.*

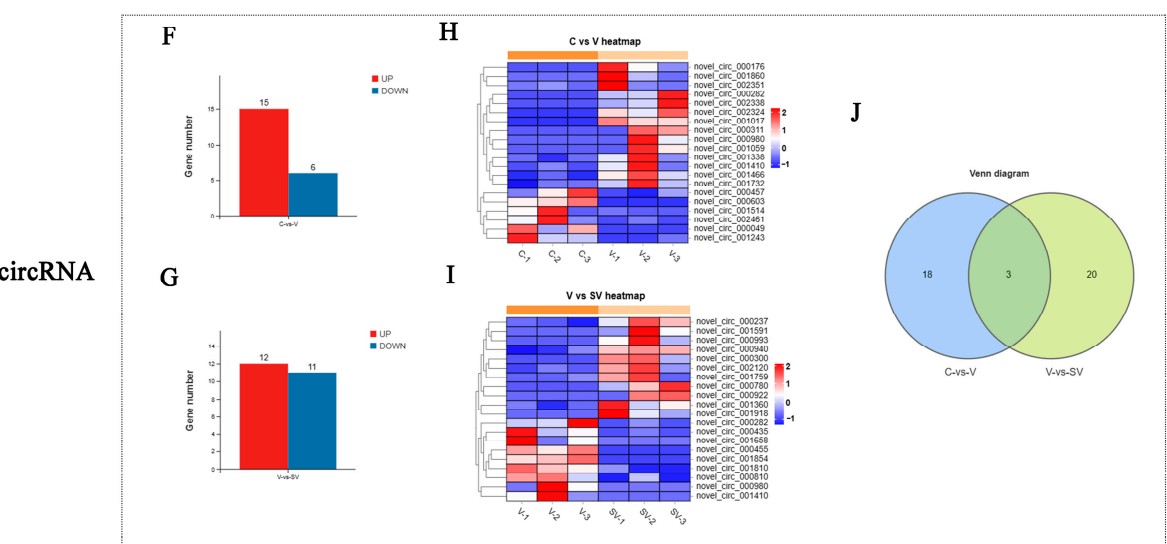

**Figure 3.** Histogram, cluster heat map and Venn chart of mRNA and LncRNA. 144 differentially expressed miRNAs (**A**) and 21 circRNAs (**F**) between C group and V group (FC ≥ 1.2, *p* < 0.05). 117 differentially expressed mRNAs (**C**) and 23 circRNAs (**H**) between V group and SV group (FC ≥ 1.2, *p* < 0.05). Heat map generated by hierarchical clustering analysis of differentially expressed top 20 miRNAs (**B,D**) circRNAs (**G,I**). 45 differentially co-expressed miRNAs (**E**) and 3 circRNAs (**J**) were shown in Venn chart.

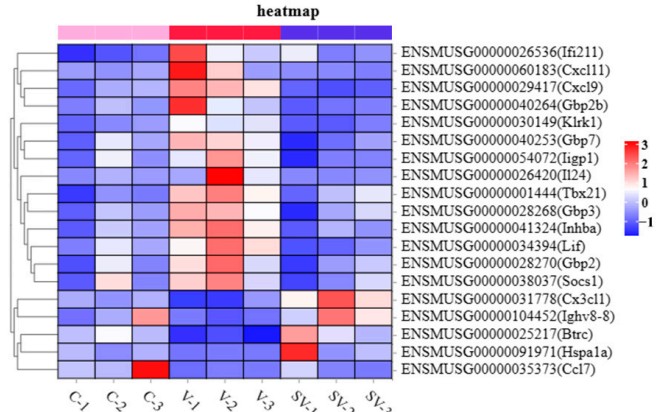

**Figure 4.** Heat map of 19 differentially co-expression mRNAs related to inflammatory responses. Red represents higher expression and blue represents lower expression.

### *3.2. GO, KEGG Analysis of DEGs*

In PCV2-infected murine splenic lymphocytes at 24 h post SSP1treatment, the differentially expressed mRNAs were enriched in the biological process (BP), which mainly focused on cell process, biological regulation, metabolism, response to external stimuli, positive and negative regulation of biological processes, signaling, immune system response, detoxification, and biological adhesion, etc. DEGs were mainly involved with binding, catalytic activity, molecular functional molecule, and transcriptional regulatory activity in molecular function (MF). In terms of cellular component (CC), differential mRNA is mainly enriched in organelles, membranes, protein-containing complexes, extracellular regions, and cell junctions (Figure 5).

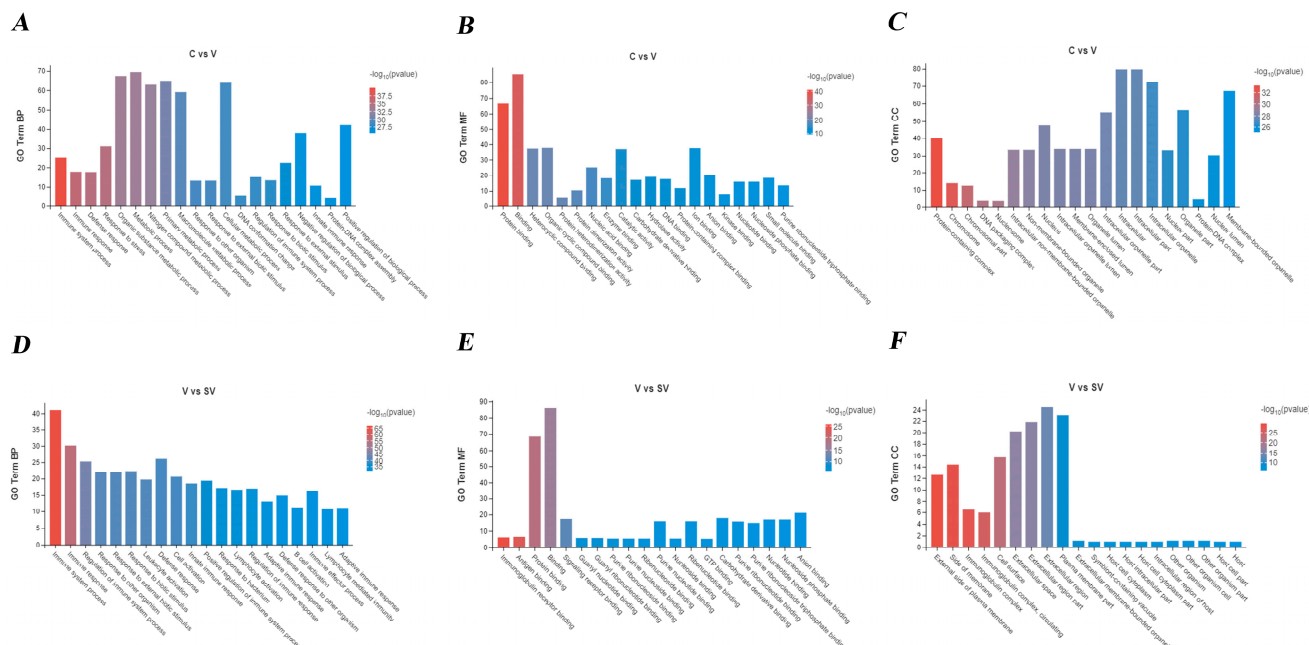

**Figure 5.** GO term analysis of DEGs. C group and V group: Gene Ontology annotation of mRNA in BP (**A**), MF (**B**), CC (**C**). V group and SV group: Top 20 enrichment genes in BP (**D**), MF (**E**), CC (**F**). GO, Gene Ontology; CC, cellular component; MF, molecular function; BP, biological process.

The KEGG shows that the differentially expressed mRNAs were enriched in the IL-17 signaling pathway, the interaction of the viral protein with cytokine and cytokine receptor, p53 signaling pathway, TNF signaling pathway, cytokine-cytokine receptor interaction, chemokine signaling pathway, C-type lectin receptor signaling pathway between the C group and the V group. DEGs between the V group and the SV group were enriched in primary immunodeficiency, natural killer cell-mediated cytotoxicity, NF-κB signaling pathway, B-cell receptor signaling pathway, FcγR-mediated phagocytosis, PI3K-Akt signaling pathway, cytokine-cytokine receptor interaction and TNF signaling pathway (Figure 6).

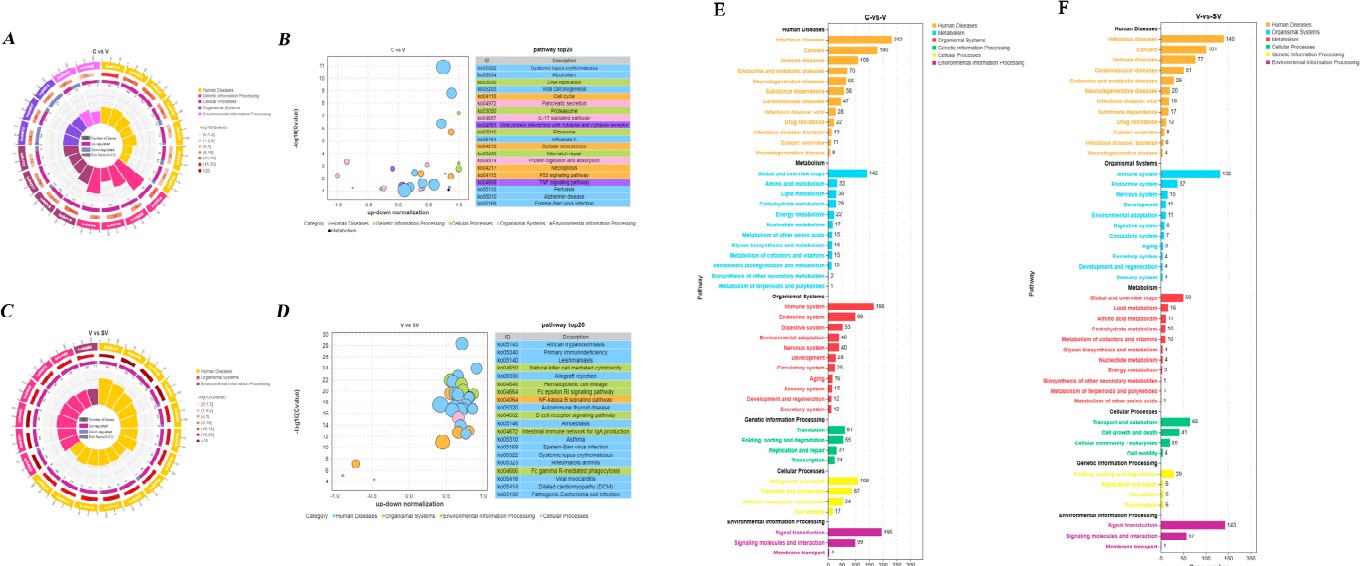

**Figure 6.** KEGG analysis of DE mRNAs. Enrichment and screening of KEGG pathway in C group and V group resulted in 325 signaling pathways, involving IL-17 signaling pathway, interaction of viral protein with cytokine and cytokine receptor, p53 signaling pathway, TNF signaling pathway,

cytokine-cytokine receptor interaction, chemokine signaling pathway, C-type lectin receptor signaling pathway (**A**,**B**,**E**). 286 signaling pathways were obtained by enrichment and screening of the KEGG pathway in the V group and the SV group, which mainly involved primary immunodeficiency, natural killer cell-mediated cytotoxicity, NF-κB signaling pathway, B-cell receptor signaling pathway, FcγR-mediated phagocytosis, PI3K-Akt signaling pathway, cytokine-cytokine receptor interaction and TNF signaling pathway (**C**,**D**,**F**).

### 3.3. Series Test of Cluster and Transcription Factor Targeting Analysis

The trend of differentially expressed mRNAs in C, V, and the SV group were analyzed, and eight trend cluster plots were obtained using the STEM algorithm (Figure 7). Transcription factor targeting analysis showed that Irf1, E2F2 and Jun were targeted by differentially expressed genes in both comparison groups (C vs V, V vs SV, Table S2).

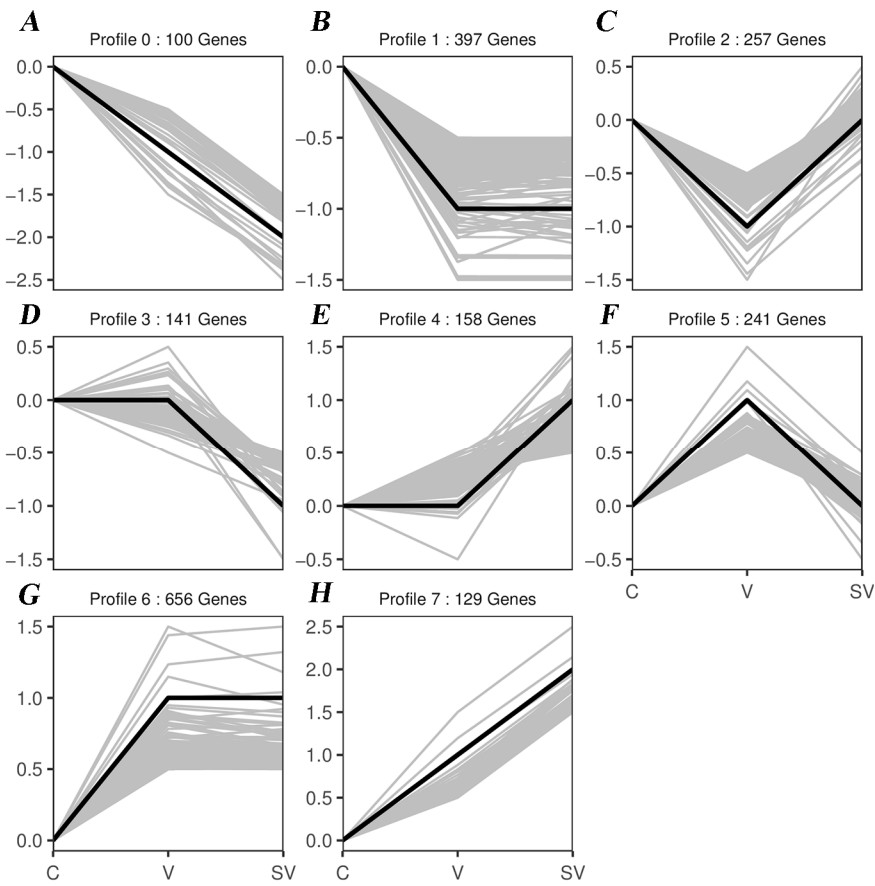

**Figure 7.** Trend cluster maps from DEGs by STEM analysis. (**A**) 100 DE mRNAs were down-regulated. (**B**) 397 DE mRNAs were down-regulated and then stabilized. (**C**) 257 DE mRNAs were down-regulated and then up-regulated. (**D**) 141 DE mRNAs were stabilized and then down-regulated. (**E**) 158 DE mRNAs were stabilized and then up-regulated. (**F**) 257 DE mRNAs were up-regulated and then down-regulated. (**G**) 686 DE mRNAs were up-regulated and then stabilized. (**H**) 129 DE mRNAs were up-regulated.

### 3.4. Validation of lncRNA, mRNA and miRNA

To verify the reliability of RNA-seq data, 18 mRNAs (CXCL10, CXCL9, IRF7, STAT1, SOCS1, SP140, FURIN, ZBP1, Serpina3g, JUN, RPS2, GBP7, GBP2, SAMHD1, GBP4, ACOD1, TNF-α, IFIT2), six lncRNAs (MSTRG.17554.2, MSTRG.4833.1, MSTRG.9203.1, MSTRG.9900.7, MSTRG.12578.2, MSTRG.4832.1) and four miRNAs (miR-187-y, miR-372-y, miR-125-z, let-7-z) were randomly selected for RT-qPCR verification (Figure 8). The results showed that the relative expression of lncRNA or mRNA by RT-qPCR was consistent with the sequencing results, as shown in Figure 7. Collectively, only two miRNAs

(miR-372-y, miR-125-z) and one lncRNA (MSTRG.9203.1) expression mismatched the RNA-seq data among the 28 detected RNAs. The above data suggested the reliability of the RNA-seq analysis.

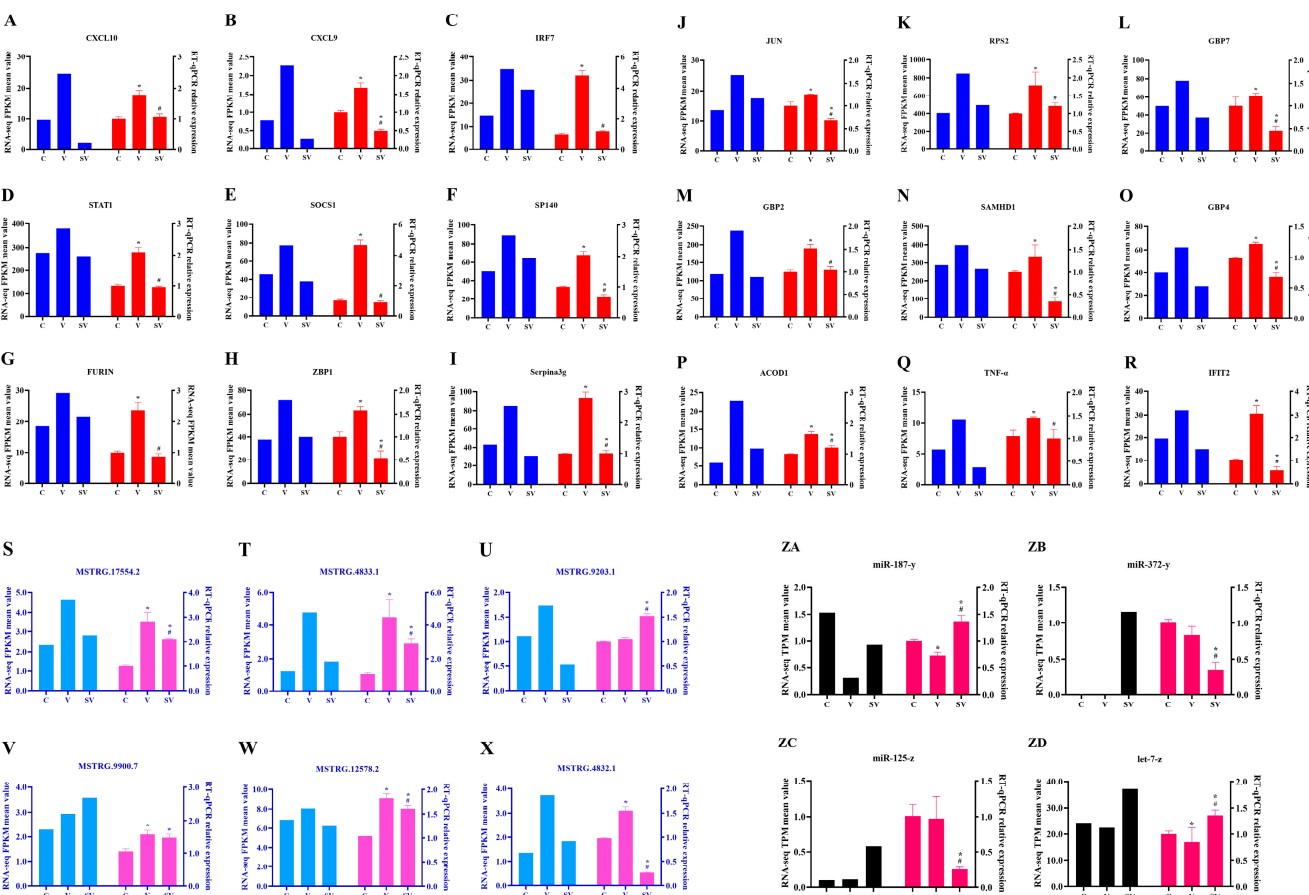

**Figure 8.** RT-qPCR validation of the expression of miRNA, lncRNA and mRNA (*n* = 3). Differentially expressed mRNA, LncRNA and miRNA were randomly chosen to validate the reliability of the RNA-seq analysis (*n* = 3). (**A–R**) The relative expression of partial mRNAs; (**S–X**) The relative expression of partial LncRNAs; (**ZA–ZD**) The relative expression of partial miRNAs. * represented significant or extremely significant difference compared with C group; # meant significant or extremely significant difference compared with V group.

### 3.5. Intervention Effect of SSP1 on Inflammation-Related Protein Expression in PCV2-Infected Splenic Lymphocytes

Based on the results of KEGG analysis, we screened the NF-κB signaling pathway, TNF signaling pathway, and NOD-like receptor signaling pathway-related proteins to explore the intervention mechanism of SSP1 on PCV2-induced inflammatory response.

The protein expression levels of p-IκB, p-p65, GBP2, TNF-α, and IRF1 were detected. As seen in Figure 9, the protein expression levels of p-p65, GBP2, TNF-α, and IRF1 increased significantly ($p < 0.01$), and the phosphorylation level of *p*-IκB-α and p-SAMHD1 were observably enhanced ($p < 0.05$) after 24 h PCV2 infection. SSP1 treatment alleviated the expression of these proteins. Meanwhile SSP1 could significantly reduce protein expression level of GBP2 ($p < 0.05$) and p-SAMHD1 ($p < 0.01$) in murine splenic lymphocytes.

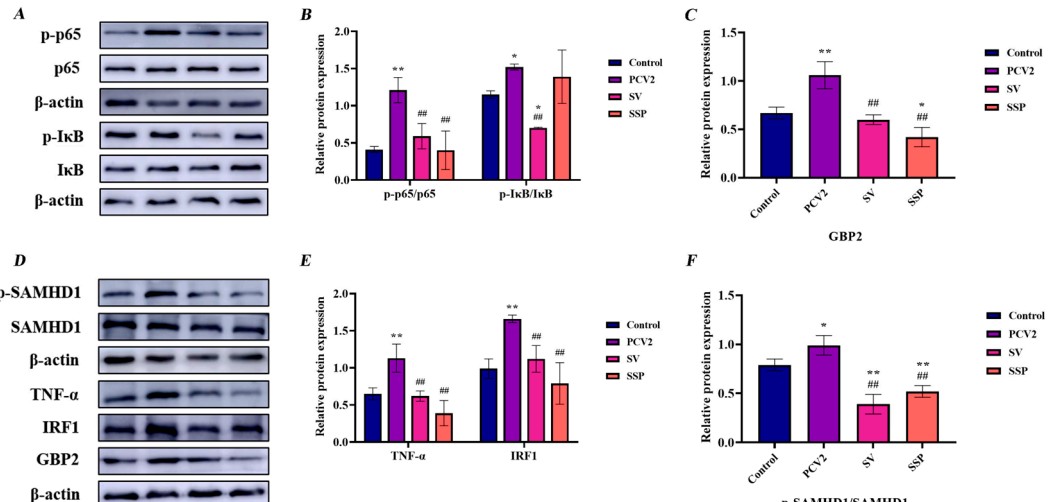

**Figure 9.** Effect of SSP1 on NF-κB/TNF/ cytosolic DNA sensing pathway in PCV2-infected murine splenic lymphocytes. (**A**) The protein bands of p-p65, p65, p-IκB and IκB. (**B**) Analysis of protein gray value of p-p65 / p65 and p-IκB / IκB. (**C**) Analysis of protein gray value of GBP2. (**D**) The protein bands of p-SAMHD1, SAMHD1, TNF-α, IRF1, and GBP2. (**E**) Analysis of protein gray value of TNF-α and IRF1. (**F**) Analysis of protein gray value of p-SAMHD1/ SAMHD1. * or ** represented significant or extremely significant difference compared with C group; ## meant significant or extremely significant difference compared with V group.

### 3.6. Target Gene Prediction and ceRNA Network Construction

Based on the miRanda analysis, the differential mRNA anti-correlated miRNA associated with 62 inflammatory genes and the differential LncRNA associated with these miRNAs were screened to construct the ceRNA network according to Pearson's correlation coefficient. These data are further combined with positively correlated LncRNA-mRNA pairs. Finally, a ceRNA network diagram was constructed. The results showed that a complex modulatory mechanism was involved in the course of PCV2 infection, but no significant interactions after SSP1 treatment in the ceRNA network, indicating that the SSP1 intervention in the PCV2 infection may not be through the ceRNA mechanism (Figure 10).

**A**

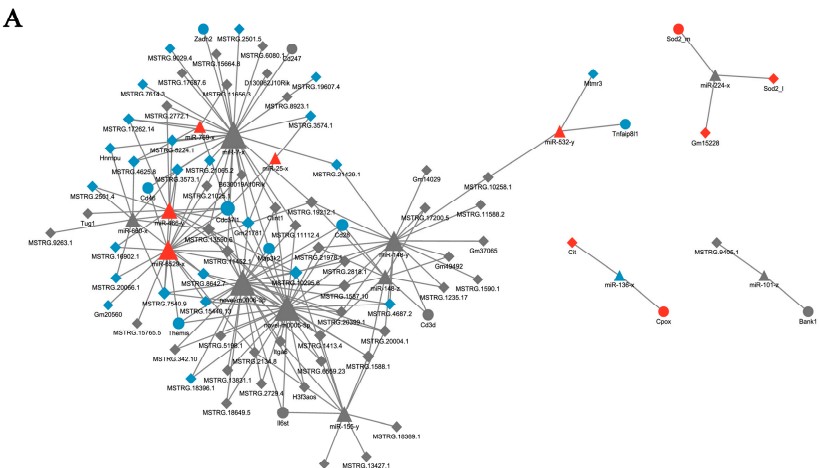

**Figure 10.** *Cont.*

**B**

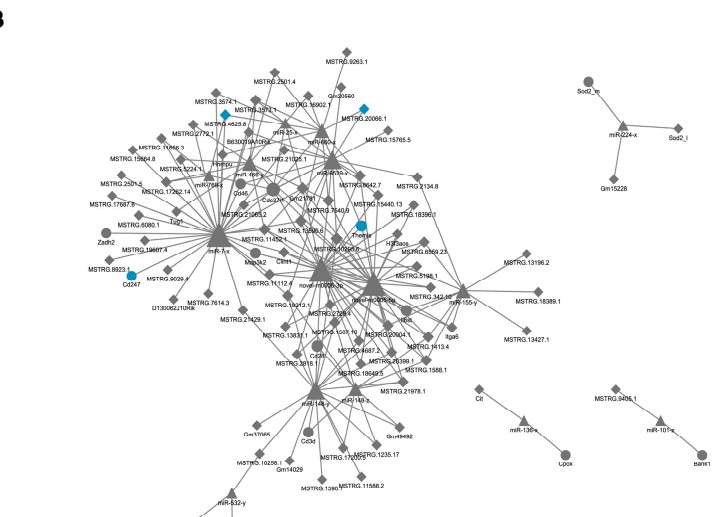

**Figure 10.** Competing endogenous RNAs (ceRNAs) network construction. Differentially expressed mRNAs, miRNAs and lncRNAs, and in the C group and V group (**A**), and in the V group and SV group (**B**) were integrated to construct the ceRNA network. The up-regulated mRNA was represented as a red circle, while the down-regulated mRNA was represented as a blue circle. The up-regulated LncRNA was represented by a red diamond, while the down-regulated lncRNA was represented by a blue diamond. The up-regulated miRNA is represented by a red triangle, while the down-regulated miRNA is represented by a blue triangle.

### 3.7. The Interaction between LncRNAs, mRNAs and miRNAs in the V and SV Group

The mRNAs related to inflammatory response were screened from 158 differentially co-expressed mRNAs between the C group, V group, and SV group, and the differential lncRNA with trans, cis, and antisense relationships and differential miRNAs with targeted relationships with mRNA were screened, and the LncRNA-miRNA-mRNA network diagram was constructed by Cytoscape (Figure 11). The results showed that miRNAs targeted with these mRNAs were mostly regulated by miR-7032-y, miR-328-y, miR-484-z, the miRNAs played an important role in SSP1 regulation of PCV2 infection, while lncRNA did little work in the regulation process, indicating that the miRNA-mRNA network may be the main approach of SSP1 to regulate inflammation induced by PCV2.

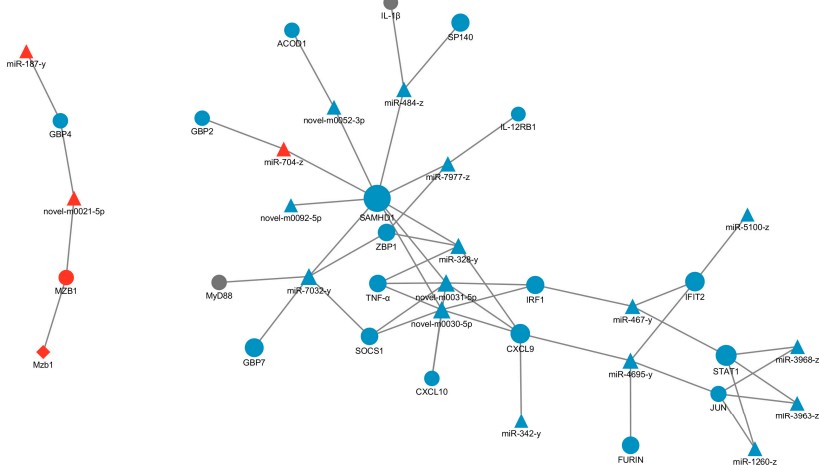

**Figure 11.** LncRNA, mRNA and miRNA interaction network diagram in V group and SV group.

## 4. Discussion

Transcriptome sequencing technology is an effective tool to explain the regulatory mechanisms of bioactive molecules [28]. SSP1, as one of the important components of *Sophora subprostrate*, has shown good anti-inflammatory and antioxidant effects [29]. Much evidence showed that *Sophora subprostrate* polysaccharides have potential applications in the prevention of chronic infectious diseases. In our previous study, PCV2-infected murine splenic lymphocytes could induce an inflammatory response, while SSP1 could modulate this inflammatory damage [12]. Therefore, to understand the underlying ncRNA mechanism of SSP1 in regulating PCV2-induced splenic lymphocyte injury in mice, the transcripts were annotated in samples using RNA-seq to analyze differentially expressed mRNA, LncRNA, miRNA and circRNA and their complex regulatory relationships, verify the reliability of differentially expressed genes by RT-qPCR and the expression of key signaling pathway proteins for DE mRNAs enrichment by western blot, which provided a basis for us to understand the regulatory mechanism of SSP1. In addition, these reactive ncRNAs are potential biomarkers and therapeutic targets, which provide a theoretical basis for the practical application of *Sophora subprostrate* polysaccharides.

### 4.1. SSP1 Alleviates PCV2-Induced Inflammation in Splenic Lymphocytes by Modulating Inflammatory Gene Responses

To elucidate the underlying molecular mechanism of the intervention of SSP1 in PCV2-induced inflammatory responses, the transcripts were identified between these three groups (C, V, SV group). The sequencing results showed that PCV2 infection can induce up-regulated expression of chemokines and inflammatory factors such as CXCL9, CXCL10, CXCL11, TNF-$\alpha$, and IL-1$\beta$, while SSP1 can downregulate the expression levels of these factors, which is consistent with previous reports [12]. In the present study, pro-inflammatory signaling molecules (IRF, STAT, SOCS) and pro-inflammatory cytokine-related genes (TNF-$\alpha$, CXCL9, CXCL10) occupy a more important position in differential expression enrichment analysis. mRNAs such as SAMDH1, IRF7, STAT1, GBP2, and JUN are also differentially expressed significantly. SAMHD1 is a host-limiting immune factor that inhibits viral infection and replication by hydrolyzing intracellular dNTPs. Previous studies have shown that SAMHD1 is not only involved in autoimmune regulation, but also TNF-$\alpha$-induced inflammatory responses [30]. Its activity is negatively regulated by phosphorylation, and when SAMHD1 is phosphorylated, its antiviral ability is weakened [31]. Stimulation with HIV-1-induced cytokines increased the high expression of the antiviral factors GBP2 and GBP5 in human cells [32]. GBP2 can directly kill intracellular pathogens and activate downstream inflammasomes after being activated by interferon (IFN), which is an important component of host defense against intracellular pathogens [30]. STAT1 is an important mediator in mediating the IFN response and plays a key role in the immune response. Macrophages mainly kill viruses by producing reactive oxygen species (ROS) and reactive nitrogen intermediates (RNI) through the IFN$\gamma$/STAT1 pathway [33]. IRF7 is a transcriptional regulator of type I IFNs and IFN-induced genes. Once phosphorylated, the activated IRF7 can enter the nucleus and bind to IFN-stimulated response element (ISRE) sites, resulting in the secretion of type I IFN and other inflammatory cytokines [34]. PCV2 infection induced an immune response in murine splenic lymphocytes in vitro, thereby causing an inflammatory response, while SSP1 inhibited the secretion of pro-inflammatory factors and mRNA expression levels caused by PCV2 to a certain extent. Based on these studies, we hypothesize that SSP1 protects cells against PCV2 infection by regulating inflammatory responses.

### 4.2. SSP1 May Attenuate the Activation of the TNF/NF-$\kappa$B Signalling Pathway through the IL12 Signalling Pathway, Thereby Reducing the Inflammatory Response

GO enrichment analysis showed that DEGs were mainly involved in bioregulation, metabolism, stimulatory response, signaling immune system, and biological adhesion, which indicated that the immune system could be mobilized to resist viral stimulation in

the process of splenic lymphocytes infected by PCV2 by SSP1 [35,36]. KEGG enrichment analysis suggested that SSP1 can improve intracellular inflammatory responses by down-regulating the expression levels of genes related to the NF-κB signaling pathway and TNF signaling pathway, which are the typical pro-inflammatory signaling pathways [37,38]. We detected these key proteins in the key signaling pathway and found that the TNF signaling pathway in splenic lymphocytes was activated after PCV2 infection, which in turn stimulated IKK phosphorylation, resulting in the activation of NF-κB signaling pathway, phosphorylation of IκB and NF-κB p65, and phosphorylation of p65 into the nucleus to trigger an inflammatory chain reaction. In addition, the activation of the TNF signaling pathway also indirectly promotes the expression of IRF1, further mediates the transcription of inflammatory factors, and activates the immune response of cells [39]. IRF1 is a transcription factor that binds to and drives the transcription of the immune defense molecules GBP5, Nos2, and CXCL10 to mediate the transcription of inflammatory genes, playing an important role in innate immunity [40]. SSP1 can reduce the protein expression levels of TNF-α and IRF1, inhibit the phosphorylation levels of IκB and NF-κB p65, thereby preventing the nucleation transcription of phosphorylated p65, and alleviate the inflammatory response induced by PCV2. In addition, SSP1 can inhibit protein expression of GBP2 and phosphorylated SAMHD1, which is consistent with the change in gene expression level. The sequencing results showed that IL-12RB1 receptor expression was significantly reduced in the SSP1 treatment group. The pro-inflammatory cytokine interleukin-12 induces expression of the receptor activator of nuclear factor-κB ligand (RANKL) in human periodontal ligament cells [41]. IL-12 was found to act directly on tumor cells to activate NF-κB and enhance IFN-γ-mediated STAT1 phosphorylation [42]. This bears some resemblance to the results of our study. The IL-12 signaling pathway was closely related to the activation of NF-κB, PI3K-Akt, and p38 MAPK signaling pathways. SSP1 may weaken the activation of pro-inflammatory signaling pathways through this pathway, thereby alleviating the inflammatory response.

### 4.3. SSP1 Regulates miRNA-mRNA Networks to Modulate PCV2-Induced Inflammatory Responses

In this study, LncRNA-miRNA-mRNA regulatory network diagram shows that DE mRNAs (SAMHD1, TNF-α, IRF1, CXCL9, STAT1, JUN) and DE miRNAs (miR-7032-y, miR-328-y, miR-484-z, and others) have an important r role in regulating PCV2-induced inflammatory responses. miR-328 and miR-484 may play roles in regulating the assembly or activation of inflammasomes. The miR-484 is a key regulator in common cancers and non-cancer diseases, targeting inflammation, apoptosis, and mitochondrial function-related mRNAs (SMAD7, Fis1, YAP1, etc.), and plays an important role in fighting disease [43]. Up-regulation of miR-484 can inhibit the PI3K/AKT signaling pathway and target the expression of transcription factor CCL-18 to inhibit cell proliferation, migration, and invasion [44]. Xu et al. investigated the role of Yap1 and miR-484 in lipopolysaccharide (LPS)-treated H9c2 cells, indicating that miR-484 inhibitor significantly improved cell viability, decreased apoptosis, suppressed NLRP3 inflammasome formation, and reduced secretion of inflammatory cytokines TNF-α, IL-1β, and IL-6 [45]. Overexpression of miR-328-3p can increase chondrocyte viability, reduce chondrocyte apoptosis levels, and alleviate the disease progression of osteoarthritis [46]. Besides, miR-328-3p overexpression can inactivate the PI3K/AKT signaling pathway and inhibit cell proliferation and metastasis in colorectal cancer [47]. Mendonça LSO evaluated the expression of circulating microRNAs related to inflammasome regulation in twenty-seven patients with cutaneous leishmaniasis, the results showed that an increase in the expression of miR-328-3p, indicating that miR-328 plays a role in regulating the assembly or activation of inflammasomes [48].

It has been shown that PCV2 regulates cellular inflammatory responses through the dysregulation of cellular miRNA-mRNA networks and that host miRNAs may be dysregulated by PCV2 infection and play an important role in the regulation of inflammation by PCV2 [25]. Multiple interacting miRNA-mRNA axes play a role in the regulation of

PCV2 infection by SSP1. In contrast, our study showed that the regulatory role of SSP1 in regulating PCV2-induced inflammation in splenic lymphocytes was also closely linked to miRNA. Therefore, we speculate that miRNA-regulated immune signaling pathways may be an important reason for the antiviral effect of SSP1. Functional studies of DE miRNAs, especially miR-7032-y, miR-328-y, and miR-484-z may provide important directions for further research on the mechanism of SSP1 regulation of PCV2 infection. In addition, due to the complexity of the LncRNA-miRNA-mRNA regulatory network, the regulatory effects between specific ncRNAs and related inflammatory response genes can be verified by overexpression/inhibition in follow-up stages. In our other research, PCV2 infection could induce inflammation in swine splenic lymphocytes, SSP1 alleviated the inflammatory response to PCV2 infection. We will compare the differences between PCV2-infected murine spleen lymphocytes and porcine spleen lymphocytes and the effect of polysaccharides on the antiviral effect of PCV2 infection in these two cell types in further exploration.

## 5. Conclusions

In this study, statistical analyses of differentially expressed transcripts were per-formed to discover the key gene in regulating inflammatory responses. SSP1 can reduce the protein expression levels of TNF-$\alpha$ and IRF1, inhibit the phosphorylation levels of I$\kappa$B and NF-$\kappa$B p65, thereby preventing the nucleation transcription of phosphorylated p65, and alleviate the inflammatory response induced by PCV2. Multiple interacting miRNA-mRNA axes play a role in the regulation of PCV2 infection by SSP1, and its interventional mechanism is mainly involved in the key differential miRNA including miR-7032-y, miR-328-y, miR-484-z. The results provided a basis for us to understand the regulatory mechanism of *Sophora subprostrate* polysaccharides, and the key ncRNAs may be potential biomarkers and therapeutic targets, which provide a theoretical basis for the practical application of *Sophora subprostrate* polysaccharides.

**Supplementary Materials:** The following supporting information can be downloaded at: https://www.mdpi.com/article/10.3390/cimb45070383/s1, Table S1: Output statistics of the sequencing reads for each sample; Table S2: Transcription factor targeting analysis; Table S3: Primer sequence of mRNA; Table S4. Primer sequence of LncRNA; Table S5. Primer sequence of miRNA.

**Author Contributions:** Conceptualization, Y.Z., N.J. and T.H.; methodology, N.J. and Q.C.; valida-tion, N.J. and X.X.; formal analysis, Y.Z. and Q.C.; investigation, Y.Z., N.J. and T.H; resources, T.H.; data curation, X.X. and Q.C.; writing—original draft preparation, Y.Z. and N.J.; writing—review and editing, Y.Z., N.J. and T.H.; visualization, X.X. and Q.C.; supervision, Y.Z.; project administration, N.J.; funding acquisition, T.H. All authors have read and agreed to the published version of the manuscript.

**Funding:** This research was funded by the National Natural Science Foundation of China, grant number 31960715.

**Institutional Review Board Statement:** The study was conducted in accordance with the Declaration of Guangxi University, and approved by the Institution Animal Ethics Committee of Guangxi University (GXU2018-009) for studies involving animals.

**Informed Consent Statement:** Not applicable.

**Data Availability Statement:** Some or all data, models, or codes generated or used during the study are available from the corresponding author by request. Gene data can be downloaded at https://ngdc.cncb.ac.cn/gsa/ (accessed on 17 April 2023), and the gene accession number is CRA010694 and CRA010695.

**Acknowledgments:** The authors are grateful to the Key Laboratory of Animal Diseases Diagnostic and Immunology of the Ministry of Agriculture at Nanjing Agricultural University for providing the PCV2 SH strain.

**Conflicts of Interest:** The authors declare no conflict of interest.

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
