# Peer review of "Whole Transcriptome Analysis of Intervention Effect of Sophora subprostrate Polysaccharide on Inflammation in PCV2 Infected Murine Splenic Lymphocytes"

_cimb, doi:10.3390/cimb45070383_

Round 1

Reviewer 1 Report

Comments and Suggestions for Authors

The objective of this paper was to identify and evaluate effect of Sophora subprostrate Polysaccharide on PCV2 infected splenic lymphocytes. Authors used advanced data processing techniques, but methodology has methodological flaws.

Major comments:

Why Authors used only PCV2 vaccine strains SH?

Why Autors used murine splenic lymphocytes instead of swine splenic lymphocytes?

Line 55,56 - Sophora subprostrate, ….anti-inflammation antioxidant and immunomodulation. Please add references

Line 115 - Murine splenic lymphocytes were obtained in accordance with the instructions – according which instruction?

Line 143 – Authors cultured lymphocytes only 6 h before PCV2 infetcion. Most often lymphocytes are growth for 12-24 h or longer,  for 4-7 days.

Line 118 - Lack of name of splenic lymphocyte isolation liquid kit and used technique. How Authors assesed lymphocytes (CD4 or CD8) purity?

Line 156 – Lack of information about used reverse transcriptase

Line 158, 159 – Then base A was added at the 3' end, and a sequencing linker was added to connect to base A. – this sentence should be expand and detailed. 

The discussion is too long and lacks focus on the topic. The authors need to revise the discussion with adequate references that presents the background and more detailed aim and conclusion of this study.

Minor comments:

Abstract – please add what is Sophora subprostrate

Line 89 - miRNA are closely related to PCV2 infection proces - Please add references and details like miR number

Line 110 – please add some details about PCV2 virus propagation eg. Numer of passage, and add PK-15 cells provider

Line 114 - mice were euthanized and dissected

Line 141 - SSP1 treatment group (SV) – it was PCV2 infected group tratment with SSP1?

Line 173 – p value instead of P

Line 188,189 – please redfrat sentecne : After cDNA was obtained by reverse transcription After RNA was reverse-transcribed to cDNA by the kit

Line 195 - total protein was isolated from cells by RIPA..

Line 201 - was performed overnight - lack of temperature

Line 202 - lack of secondary antibodies dilution

Line 397-401 – sentence about HIV and IFN does not match to article

Lin 408 - in vitro

Please unify in text - TNF-α or TNF

In text Author add section Reagents, but lack of lot. Please add details of reagents producer: Sephadex G 100, dNTPs, RNaseH, and DNaseI, random hexamers, QiaQuick PCR kit, Uracil-N-glycosylase enzyme, cDNA kit (line 189), 10% SDS-PAGE (if precast), skim-milk, ECL chromogenic substrate, TBST

Please add producer for: Nanodrop, Agilent2100, STEM software, JASPER database, MEME FIMO software, RT-qPCR machine (lack of model and producer), vertical electrophoresis tank (lack of model and producer), WB transfer unit (lack of model and producer), chemiluminescence detection system (lack of model and producer), Image J software.

Reviewer 2 Report

Comments and Suggestions for Authors

Dear Authors,

The work concerns the study of the regulation mechanism of Sophora subprostate polysaccharide, SSP1, in the reaction of splenocyte inflammation. The work is well written, but needs some minor corrections:

- keywords should be listed alphabetically

- the reaction between lines 107 and 108 needs to be represented in the form of a Figure 1.

- please include in the Manuscript the section on cell culture: after how many passages of murine splenic lymphocytes were used for research? Were they supplemented with antibiotics? if so, please specify their unit in which they were used.

- line 161 - please specify the conditions of the PCR reaction, taking into account the composition of the reaction mixture

- line 184 - please specify the conditions of the RT-qPCR reaction

- line 202 - please provide secondary antibodies dilution. What were the antibodies?

- Please add to the Conclusions what benefits come from the experiments performed and how they can be used in future research.

Best regards

Round 2

Reviewer 1 Report

Comments and Suggestions for Authors

I am glad that the Authors added more details in methods. Moreover, the authors followed my suggestions, answered my questions. 

If Authors used both PCV2 SH strain and PCV2 (PCV2NJ2002) isolated from a kidney tissue, and both strains produced similar effects during the experiment, it should be added to Methods section. Moreover, if Authors, in preliminary study, infected on both porcine spleen lymphocytes and murine spleen lymphocytes it should be also added to Methods section.

Same, if the CD4+ and CD8+ purity of lymphocytes was assesed by flow cytometry, it should be added to Methods section.
